# The High Flux of Superhydrophilic-Superhydrophobic Janus Membrane of cPVA-PVDF/PMMA/GO by Layer-by-Layer Electrospinning for High Efficiency Oil-Water Separation

**DOI:** 10.3390/polym14030621

**Published:** 2022-02-05

**Authors:** Han Wu, Jia Shi, Xin Ning, Yun-Ze Long, Jie Zheng

**Affiliations:** 1Industrial Research Institute of Nonwovens & Technical Textiles, College of Textiles & Clothing, Shandong Center for Engineered Nonwovens, Qingdao University, Qingdao 266071, China; whqddx@126.com (H.W.); s1109907102@163.com (J.S.); xning_irintt@163.com (X.N.); yunze.long@163.com (Y.-Z.L.); 2Collaborative Innovation Center for Nanomaterials & Devices, College of Physics, Qingdao University, Qingdao 266071, China

**Keywords:** oil-water separation, Janus membrane, electrospun nanofiber, superhydrophobic

## Abstract

A simple and novel strategy of superhydrophilic-superhydrophobic Janus membrane was provided here to deal with the increasingly serious oil-water separation problem, which has a very bad impact on environmental pollution and resource recycling. The Janus membrane of cPVA-PVDF/PMMA/GO with opposite hydrophilic and hydrophobic properties was prepared by layer-by-layer electrospinning. The structure of the Janus membrane is as follows: firstly, the mixed solution of polyvinylidene fluoride (PVDF), polymethylmethacrylate (PMMA) and graphene oxide (GO) was electrospun to form a hydrophobic layer, then polyvinyl alcohol (PVA) nanofiber was coated onto the hydrophobic membrane by layer-by-layer electrospinning to form a composite membrane, and finally, the composite membrane was crosslinked to obtain a Janus membrane. The addition of GO can significantly improve the hydrophobicity, mechanical strength and stability of the Janus membrane. In addition, the prepared Janus membrane still maintained good oil-water separation performance and its separation efficiency almost did not decrease after many oil-water separation experiments. The flux in the process of oil-water separation can reach 1909.9 L m^−2^ h^−1^, and the separation efficiency can reach 99.9%. This not only proves the separation effect of the nanocomposite membrane, but also shows its high stability and recyclability. The asymmetric Janus membrane shows good oil-water selectivity, which gives Janus membrane broad application prospects in many fields.

## 1. Introduction

The frequent occurrence of oil spills and the discharge of oily wastewater have made the problem of oil-water separation increasingly serious. The low efficiency of oil-water separation, poor circulation stability and high cost limit requires industrial and social development. Electrospinning is a simple and widely used technology in the formation of micro-nanofibers. It can easily and effectively prepare one-dimensional (1D), two-dimensional (2D) and three-dimensional (3D) electrospun nanofibers based on various materials, such as polymers, ceramics, composites, carbon materials and so on [1,2,3,4]. Electrospinning can produce nanofibers with special structures with low cost and relatively high productivity, especially ultra-fine fiber materials with large specific surface area and long length; it has broad prospects in many fields, such as immunoassay, catalysis, tissue engineering, sensors, composite reinforcement, adsorption and filtration, etc. [5,6,7,8,9].

The name Janus membrane comes from Janus, the two-faced god in ancient Roman mythology. The deity has two faces, facing opposite directions, from which the Janus membrane gets its name. The concept of Janus particle was first proposed by P.G. de Gennes in his Nobel speech in 1991 [10]. Janus membrane is a general term for membranes with opposite properties, such as composition, morphology, wettability, surface charge, and so on. The opposite properties refer to hydrophilic/hydrophobic or positive/negative charge properties, which can be realized by chemical or physical methods. Janus membrane is specially defined by surface properties rather than components. The key to distinguishing the Janus membrane from the general asymmetric membrane is whether the properties of both sides of the membrane are opposite [11,12,13]. The biggest advantage of the Janus membrane is that it can meet the needs of two contradictory properties in a certain application at the same time. This concept has great applications in many fields, such as oil-water separation, one-way ion channel, fog collection, contact enzyme membrane reactor, solar distillation, etc. [14,15,16,17,18].

Asymmetric manufacturing is the simplest way to obtain a Janus membrane, which is to make each side of the membrane separately and then combine them together to form a Janus membrane. This method can be combined with electrospinning technology. By layer-by-layer electrospinning, two layers of nanofiber membranes with opposite properties can be combined. For example, a hydrophobic nanofiber membrane can be electrospun onto a hydrophilic nanofiber membrane. With the development of the Janus membrane, various technologies were applied to produce Janus membrane, such as electrospinning, spray modification, etc. Electrospinning is a simple and efficient method to produce a Janus membrane. By layer-by-layer electrospinning, two layers of fiber membrane with opposite properties can be combined. With the continuous research in this field and the emergence of many excellent progresses, the technology is constantly improved. In recent years, more and more Janus membranes have appeared [13,16,19].

Two-layer composite is a common method to produce Janus membrane. By electrospinning hydrophobic nanofibers onto hydrophilic fibers, two layers of Janus nanofiber membrane can be formed by layer-by-layer electrospinning. Conversely, hydrophilic nanofibers can also be electrospun onto hydrophobic fibers. When Janus membrane is produced by this method, a variety of polymers can be selected, including polyurethane (PU) [14,16,20,21,22], polyvinylidene fluoride (PVDF) [19,23,24,25,26,27,28,29,30,31,32,33], polymethyl methacrylate (PMMA) [34,35,36,37], polylactic acid (PLA) [38,39], polystyrene (PS) [26,40], polyvinyl alcohol (PVA) [21,41], cellulose acetate (CA) [22,23,33,40,42], polyacrylonitrile (PAN) [16,21,23,26,34,40,43,44,45], etc. At the same time, by adding hydrophilic and hydrophobic modifiers to the electrospinning solution, its properties can also be changed.

In this paper, a simple and convenient method was designed to produce Janus membrane. Hydrophobic modified PVDF fiber membrane was produced by electrospinning technology. Then PVA electrospinning was applied to cover the hydrophobic layer, and the composite membrane was crosslinked to obtain Janus membrane. Through this method, the mechanical properties of the prepared fiber membrane are greatly improved, and the hydrophilicity and hydrophobicity of the two sides are greatly different. The spinnability of PVDF was increased by adding PMMA. At the same time, graphene oxide (GO) was added to further modify the PVDF/PMMA fiber membrane. The experimental results show that a very small amount of GO has a great impact on the hydrophobicity of the hydrophobic layer. At the same time, the crosslinking of PVA eliminates the problem of PVA membrane dissolving in water. The hydrophobic side of the fiber membrane forms a superhydrophobic interface and the difference in hydrophilicity and hydrophobicity between the two sides of the Janus membrane increases. A superhydrophobic Janus membrane with high flux, recyclability and high separation efficiency was obtained. The asymmetry of the two surfaces makes the Janus membrane promising in oil-water separation.

## 2. Materials and Methods

### 2.1. Chemicals

Polyvinyl alcohol 1788 (PVA 1788, Mw = 84, 000–89, 000) were purchased from Aladdin, Ltd. (Shanghai, China), PVDF (Mw ≈ 1,000,000) were purchased from Arkema, Ltd. (Paris, France), PMMA (Mw ≈ 210,000) were purchased from Shunjie Plastic Technology Co., Ltd. (Dongguan, China), GO dispersion (1 mg mL^−1^) were purchased from XFNANO Materials Technology Co., Ltd. (Nanjing, China), potassium dihydrogen phosphate (KH_2_PO_4_, Mw = 136.09 gmol^−1^) were obtained from Aladdin, Ltd. (Shanghai, China), Span80 were purchased from Samchun Chemical Co., Ltd. (Seoul, Korea), hydrochloric acid (HCl, 37 wt% in H_2_O, Mw ≈ 36.46 gmol^−1^), ethanol (CH_3_CH_2_OH, Mw = 46.07 gmol^−1^), n-hexane (C_6_H_14_, Mw = 86.18 gmol^−1^), glutaraldehyde (GA, 50 wt% in H_2_O, Mw = 100.11 gmol^−1^), N, N-dimethylformamide (DMF, Mw = 73.09 gmol^−1^) and acetone (C_3_H_6_O, Mw = 58.08 gmol^−1^) were obtained from Sinopharm Chemical Reagent Co., Ltd. (Beijing, China). All the reagents are analytically pure, purchased from Aladdin, without any impurities and need no further purification.

### 2.2. Preparation of Janus Composite Membrane

A conventional electrospinning device was used in this work, which was composed of three main parts including the propulsion device, high voltage power supply and collector. Initially, 1.2 g PVDF and 0.3 g PMMA were dissolved in a mixture of 6.8 g DMF and 1.7 g acetone. Then GO dispersion with the mass fraction of 0.8%, 1% and 1.2% was added, and magnetic stirring was used at room temperature for 4 h to dissolve and mix uniformly. Then, the PVDF/PMMA/GO solution was obtained. At the same time, PVA solution was prepared—1.2 g PVA was dissolved in 8.8 g deionized water to obtain 12 wt% PVA solution, and magnetically stirred at 70 °C for 4 h to fully dissolve it. The schematic diagram of the Janus membrane preparation process is shown in Figure 1a. The PVDF/PMMA/GO solution was electrospun at a working voltage of 22.5 kV, the spinning rate was 0.5 mL h^−1^, the distance from the needle to the collector is 11 cm, and the spinning time was 3 h to obtain the PVDF/PMMA/GO fiber membrane. Then the PVDF/PMMA/GO fiber membrane was used as the collector in the preparation of the PVA fiber membrane by layer-by-layer electrospinning. The working voltage was 20 kV, the spinning rate was 0.66 mL h^−1^, the distance from the needle to the collector was 12 cm and the spinning time was 3 h to form a composite membrane. Then the PVA fiber membrane was coated onto the PVDF/PMMA/GO fiber membrane. The obtained membrane was dried in an oven at 60 °C for 12 h. The detailed process of cross-linked PVA (cPVA) is shown in Figure 1b. After that, the membrane was immersed in the crosslinking solution with GA as the crosslinking agent and hydrochloric acid as the crosslinking catalyst for 10 min [37]. Then the membrane was washed several times with 1% potassium dihydrogen phosphate solution, and dried at room temperature for 24 h. Finally, the Janus membrane was obtained.

### 2.3. Preparation and Separation of Water-in-Oil Emulsions

A certain proportion of oil and water were mixed with emulsifiers to prepare emulsion solutions. For water-in-oil emulsion, 1 g Span80 were added to 114 mL n-hexane containing 1 mL water, stirring the mixture for 3 h. Additionally, for water-in-diesel emulsion, 20 mg span80 was first dissolved in 99 mL diesel and then 1 mL water was added. The mixtures were sonicated for 1 h. The physical parameters of the water-in-diesel emulsions are shown in Table 1. The liquid (water or oil) flux through the Janus membrane is calculated by the following Equation [44]:F = V/(s × ∆T)(1)
where F (L m^−2^h^−1^) is the liquid flux, V (L) is the volume of filtrates, S (m^2^) is the effective separation area of the membrane and ∆T (h) is the separation time.

The separation efficiency (E) is calculated by the following formula [46]:E = (m/m_0_) × 100%(2)
where m is the mass of the oil obtained after separation and m_0_ is the mass of the oil in the solution before separation.

### 2.4. Characterization

Scanning electron microscope (SEM, Phenom ProX, Thermo Fisher Scientific (Waltham, MA, USA)) and transmission electron microscope (TEM, JSM-2100Plus, JEOL (Seoul, Japan)) were used to observe the morphological structure of the prepared nanofiber membrane. The chemical compositions and functional groups of the membrane nanocomposite were perceived through Fourier-transform infrared (FTIR) spectrum (Thermo Scientific Nicolet iS5, Thermo Fisher Scientific (Waltham, MA, USA)). The mechanical strength of the membrane was measured by an electronic universal material testing machine (Instron 3300, Instron (Norwood, MA, USA)). The pore size distribution of the membrane was measured by a pore size analyzer (PSM165, Topas (Frankfurt, Germany)). The wettability of the membrane surface was measured with a contact angle goniometer (JY-PHb, Chengde Jinhe Instrument Manufacturing Co., Ltd. (Chengde, China)), and the underwater oil contact angle, the water contact angle under oil, the water contact angle in the air (WCA) and the oil contact angle in the air (OCA) were measured. The contact angle measurement was repeated 10 times for each sample, and then the average value was taken. The physical parameters of the water-in-diesel emulsions were analyzed using the trace water molecule tester (ZTWS2000, Weifang Zhongte Electronic Instrument Co., Ltd. (Weifang, China)), viscometer (NDJ-1S, Shanghai Pingxuan Scientific Instrument Co., Ltd. (Shanghai, China)), and a surface tension meter (K100, Kruss Germany (Hamburg, Germany)). UV visible spectrophotometer (UV 752, Shanghai Jinghua Technology Instrument Co., Ltd. (Shanghai, China)) was used to measure the UV-Vis absorption of the liquid before and after separation. The filtration performance of the fiber membrane was tested by the filter material test bench (AFC 131, Topas (Frankfurt, Germany)).

## 3. Results and Discussion

### 3.1. Characterization of Janus Composite Membrane

The schematic diagram of the Janus membrane preparation process is shown in Figure 1a. Firstly, PVDF/PMMA/GO spinning solution was electrospun by the electrospinning technology to obtain PVDF/PMMA/GO fiber membrane. Then the PVA fiber membrane is coated onto the former fiber membrane by layer-by-layer electrospinning to form a composite membrane. Subsequently, chemical cross-linking is carried out in GA. PVA hydroxyl and GA bifunctional aldehyde molecules form acetal bonds during the cross-linking process, which improves its mechanical properties and water resistance [47]. The morphological characteristics of the two active surfaces of the Janus membrane were characterized by SEM, as shown in Figure 2a–h. After the nanofiber membrane was crosslinked, it was observed that the adjacent fibers were combined and twined, and the original straight fibers formed inter-fiber bonds at the intersection (Figure 2a,b) [41,46]. At the same time, through the analysis of the statistical distribution of fiber thickness, it can be seen that with the completion of crosslinking, the fiber becomes thicker. Figure 2c shows the SEM image of the pure PVDF fiber membrane. Compared with Figure 2c, the thickness of the PVDF/PMMA fiber membrane becomes uniform and spinnability is improved (Figure 2d). Figure 2e–g are the images after adding 0.8%, 1% and 1.2% GO dispersions, respectively. It can be clearly seen that with the addition of GO dispersions, the morphology of the fiber has changed, as shown in Figure 2e–g, the surface of the fiber is rougher, and a lot of small particles appear on the surface. However, when the concentration of GO dispersion reached 1.2%, obvious GO agglomeration appeared on the fiber surface. At the same time, the statistical distribution of fiber thickness was analyzed. It was found that the addition of PMMA has little effect on the diameter of the fibers. However, after adding GO, the non-uniformity of fibers increased, and when 1% GO was added, the uniformity of fibers was the best. Thus, only adding 1% GO dispersions in the following section is the best option to prepare PVDF/PMMA/GO hydrophobic side. It can also be proved from the cross-section (marked in red dotted line in Figure 2h) that the two membranes are compounded together, and the hydrophilic cPVA nanofibers cover the hydrophobic layer PVDF/PMMA/GO to form an interwoven nanofiber layer with a reduced pore size (Figure 2h). The TEM images of the PVDF/PMMA/GO hydrophobic side of nanofibers from Figure 2f are shown in Figure 2i. From the protruding part of the nanofiber surface, it can be seen that GO was dispersed on the surface and inside the nanofiber, greatly improving the fiber mechanical properties and hydrophobicity, which will be further discussed in Section 3.2 and Section 3.3.

FTIR characterizes the chemical groups of the Janus composite membranes. Figure 3a shows the FTIR spectrum of the hydrophilic side. The peak between 3550 and 3220 cm^−1^ is related to the stretching of the O-H band connected by intermolecular and intramolecular hydrogen bonds. The intensity of the O-H peak of cPVA is greatly reduced compared with that of PVA. This is because an acetal bridge or hemiacetal bridge is formed in the hydroxyl group of the PVA nanofiber, which also indicates the success of the cross-linking. In addition, the intensity of the band between 2730 and 2860 cm^−1^ has also increased. This is because a small amount of GA may be retained after washing, resulting in O=C-H stretching vibration from the C-H of the alkyl group and the aldehyde group. The band tensile strength near 1721 cm^−1^ indicates the presence of C=O in the carbonyl group. The gradual broadening of the peak observed between 1000 and 1140 cm^−1^ is attributed to the O-C-O vibration of the acetal group. It can be seen from Figure 3b that the obvious absorption peak at 1400–1410 cm^−1^ is the bending vibration peak of CH_2_ in the PVDF structure, and 1170–1190 cm^−1^ is the stretching vibration peak of CF_2_ in PVDF. In addition, compared with PVDF, a new peak appears at 1727 cm^−1^, which is the -COOR stretching vibration peak of PMMA, which proves the success of blending. Additionally, the peaks at 880 and 836 cm^−1^ are the characteristic peaks of -CH_2_-CF_2_-. Because the mass fraction of GO dispersion solution in the total solution is only 1%, no obvious peak increase was observed on FTIR. FTIR proved the successful cross-linking of PVA fiber, and also proved the successful composite of PVDF and PMMA [41,47,48,49,50].

### 3.2. Mechanical Properties of Janus Composite Membrane

Here, we prepared a kind of Janus membrane with excellent mechanical properties. In order to test the tensile properties of the fiber membrane, the Janus composite membrane was tested by a material testing machine. The maximum tensile stress and ultimate strain of PVA electrospun fiber membrane are 6.35 MPa and 108.38%, respectively, as shown in Figure 4a. After crosslinking, the maximum tensile stress of the cPVA membrane increased to about three times of the original data, reaching 20.21 MPa, and the tensile strain decreased to 50.72% (Figure 4a). The increase of the maximum tensile stress indicates that the mechanical properties of the PVA nanofiber membrane are significantly improved by crosslinking. Therefore, the crosslinking of PVA greatly improves the mechanical properties of the material. As shown in Figure 4b, it can be clearly seen that compared with pure PVDF, the elongation at break of the fiber membrane is greatly improved after adding PMMA, which can be attributed to the decrease of crystallinity of PVDF and the decrease of the brittleness. At the same time, the tensile stress of the composite membrane is also improved after adding PMMA [51]. On this basis, the mechanical properties of the fiber membrane with 1% GO dispersions were tested. It can be seen that the elongation at break and tensile stress are improved. The addition of GO improves the surface roughness of the fiber, increases the friction between fibers and makes fiber fracture difficult, thus improving the mechanical properties of the nanofibers.

### 3.3. Wettability of Janus Composite Membrane

The interfacial properties of Janus nanocomposite membranes were studied by measuring the water contact angle (WCA). The porosity of the hydrophobic layer was measured to study the effect of PMMA and GO on hydrophobicity, as shown in Figure 4c. Figure 4c clearly shows that the effective pore size of the PVDF/PMMA composite membrane decreases compared with the original membrane. When GO was added, the pore size was smaller than that of the PVDF/PMMA composite membrane. The smaller pore size represents the smaller gap between fibers. The small gap has a positive effect on the improvement of hydrophobicity, which shows that after the composite is successful, it is helpful to improve the hydrophobicity of the fiber membrane. The addition of PMMA reduced the pore size of the fiber, but decreased the water contact angle, which can be attributed to the presence of ester hydrophilic groups in PMMA, which further confirms that the peak of 1727 cm^−1^ in Figure 3b is the characteristic peak of the -COOR. However, interestingly, the contact angle of water is enhanced to more than 150° after the addition of GO. It can be seen that the addition of a small amount of GO has a great influence on hydrophobicity (Figure 4d).

The surface wettability of the two active surfaces of the Janus membrane is shown in Figure 5a–h and Table 2. The hydrophilicity of the cPVA membrane was measured. Although the hydroxyl functional groups were greatly reduced due to the formation of acetal bond in the crosslinking process, the hydrophilicity of cPVA membrane remained and it did not dissolve in water, which inhibited the water solubility. Figure 5 and Table 2 show the wettability of hydrophilic cPVA and hydrophobic PVDF/PMMA/GO, which indicates that there is a significant difference in the surface wettability between the two sides of the composite membrane. The water droplets can be absorbed instantaneously on the cPVA interface, and the WCA of cPVA rapidly changes from 34 to 0° within 3 s, indicating that the cPVA has strong hydrophilicity and excellent performance as a hydrophilic carrier on the hydrophilic side (Figure 5a). In contrast, the WCA of the hydrophobic side surface of the Janus membrane is 153 ± 2.5° (Figure 5e), showing superhydrophobicity in air. In contrast, the WCA of PVDF/PMMA membrane is only 136 ± 3° (Figure 4d). Both sides of Janus membranes are lipophilic in air (Figure 5b,f). In order to further study the wettability of Janus membrane in the process of oil-water filtration, the wettability of Janus membrane under water and oil was also measured in this work. The membranes on both sides of the Janus membrane show hydrophobicity under oil (Figure 5c,g). This is because a large number of oil droplets will adhere to the surface of the oleophilic membrane, which will eventually block the hydrophobic membrane pores and prevent water from entering. The cPVA membrane is an underwater oil repellent membrane, and the underwater OCA is 145 ± 4° (Figure 5d). When the oil droplets try to wet the hydrophilic surface, it will produce a strong repulsive force against water and hinder the entry of oil droplets. In contrast, the PVDF/PMMA/GO membrane is an underwater lipophilic membrane, and the underwater OCA is 90 ± 3° (Figure 5h). In the next few seconds, the underwater OCA becomes 0° (Figure 5i), which can be attributed to the strong hydrophobic-hydrophobic interaction between the low surface energy membrane material and the oil droplets [52]. Here, we also list the results of the contact angle test of other similar works in Table 3 to compare with this work. We found that the prepared cPVA-PVDF/PMMA/GO Janus membranes by layer-by-layer electrospinning have an obvious large contact angle, which proves the superhydrophobicity of the prepared material. The contact angle test further proved the potential of Janus membrane in oil-water separation.

The addition of GO significantly reduces the surface energy of the sample, thus improving the hydrophobicity of the fiber membrane. Through the contact angle test, it is found that the PVDF/PMMA/GO layer has very low adhesion (Figure 6a), which further proves its strong hydrophobicity. In addition, Figure 6b shows the superior mechanical flexibility of the Janus membrane, which will not break after bending and folding, and can still be restored to the original state, showing that the prepared Janus membrane has excellent mechanical properties.

### 3.4. Oil-Water Separation Performance of Janus Membrane

In this work, the two sides of the Janus membrane show superhydrophilicity and superhydrophobicity, which provide a good basis for oil-water separation. The prepared cPVA-PVDF/PMMA/GO Janus membrane was placed on a filtration device to study the oil-water separation performance, as shown in Figure 7a. Its performance was tested with n-hexane and deionized water; pouring the oil-water mixed solution into the funnel through the prepared Janus membrane. After a period of time, it can be clearly seen that all the oil flows through the Janus membrane into the beaker below. On the contrary, all of the water solution is blocked above the fiber membrane by the composite membrane (Figure 7b). This proves that it has a broad prospect in oil-water separation. In addition, the separation efficiency and durability of the Janus membrane were tested. After many oil-water separation tests, it can be seen that the Janus membrane still maintains good oil-water separation performance, and the quality of n-hexane has almost not reduced (Figure 7c). The water flux of the Janus membrane is calculated by Equation (1) and the results are shown in Figure 7d, and the flux in the process of oil-water separation can reach 1909.9 L m^−2^ h^−1^. Figure 7e shows the change of separation efficiency with the increase of separation times. It can be seen that the separation efficiency of the Janus membrane has almost not changed. The experimental results show that the Janus membrane has stable performance and high separation efficiency after several cycles. At the same time, the separation performance of the Janus membrane was further tested with water-in-oil emulsion and water-in-diesel emulsion, as shown in Figure 7f–i. The separation results can be clearly seen by optical microscope photographs, indicating that the Janus membrane can effectively separate the water-in-oil emulsion and water-in-diesel emulsion. These results not only proved the separation effect of the prepared Janus membrane, but also showed the stability of its structure, forming a highly stable and reusable membrane. By testing its absorption peaks every 50 nm wavelength by UV spectrophotometer, it is found that the spectral absorption peaks of deionized water and separated water hardly change at different wavelengths, which proves that the separation of the Janus composite membrane will not cause pollution to it. Figure 7j and k shows SEM images of the dried hydrophilic layer and the hydrophobic layer of Janus membrane after separation, and it was found that the morphology of fibers did not change. Additionally, Figure 7l depicts the FTIR spectral data of deionized water and water separated by Janus composite membrane. In the water samples before and after separation, the strong peak at 3220–3550 cm^−2^ indicates the presence of the O-H group. Through comparison, it can be concluded that the Janus membrane has no pollution in the separation process [59]. Here, we also list the results of other similar works in Table 4 to compare with this work. We found that the prepared cPVA-PVDF/PMMA/GO Janus membranes by layer-by-layer electrospinning have obvious advantages in flux and separation efficiency. In this way, this work provides a simple and novel strategy to solve the problem of oil-water separation.

### 3.5. Filtration Performance Evaluation

In addition to high oil-water separation efficiency, the prepared fiber membrane also has an excellent performance in air filtration. In order to quantitatively characterize the filtration performance of the Janus fiber membrane, we systematically studied the removal efficiency and pressure drop of air particles with different particle sizes. The filtration efficiency and pressure drop of prepared Janus membranes with different basic weights were systematically tested by NaCl, as shown in Figure 8a–c. Here, we found that the hydrophobic layer had no significant effect on the filtration efficiency, thus we fixed the spinning time of the hydrophobic layer to 30 min. On this basis, the spinning time of the hydrophilic layer was adjusted to study its effect on filtration performance. Figure 8a shows the filtration efficiency curves of different hydrophilic layer spinning times (15–60 min), which shows that the filtration efficiency increases with the increase of particle size due to its limited pore size. Figure 8b shows the relationship between spinning time and basic weight as well as pressure drop. With the increase of spinning time, the basic weight of the fiber membrane shows an obvious growth trend. Similarly, the pressure drop is also proportional to the basic weight of the Janus membrane. The filtration efficiency and pressure drop are closely related to the membrane thickness, as shown in Figure 8c. The larger the basic weight is, the thicker the fiber membrane is, which leads to the greater contact probability between the particles and the fiber membrane, and the easier it is to be intercepted by the fiber membrane. At the same time, the prepared cPVA-PVDF/PMMA/GO Janus membrane has a fine diameter and large specific surface area, which can more effectively intercept particles. The filtration efficiency of the composite membrane reaches 96.7% when the basic weight reaches 2.16 g m^−2^, which is obviously higher than 80.2% when the basic weight is 1.24 g m^−2^. At the same time, the pressure drop of the fiber membrane is also at a low level, only 45 Pa. These results indicate that the Janus membrane has great potential infiltration, which further expands the potential application of the prepared cPVA-PVDF/PMMA/GO Janus membrane.

## 4. Conclusions

In conclusion, we have demonstrated a simple method to prepare cPVA-PVDF/PMMA/GO Janus membranes with different hydrophilicity and hydrophobicity for oil-water separation by layer-by-layer electrospinning. The mechanical properties, pore size distribution and contact angle of the Janus membranes prepared by adding GO dispersion were significantly improved, which further provided a superhydrophobic interface for the prepared Janus membranes. The existence of the superhydrophobic interface makes the hydrophilicity difference between the two sides larger, and the oil-water separation efficiency is higher. The prepared Janus membrane has high flux, high separation efficiency and good cycle stability, and has good application in oil-water separation. In addition, the Janus fiber membrane prepared by simple layer-by-layer electrospinning is also good in air filtration, which further expands the potential application of the prepared Janus membranes.

## Figures and Tables

**Figure 1 polymers-14-00621-f001:**
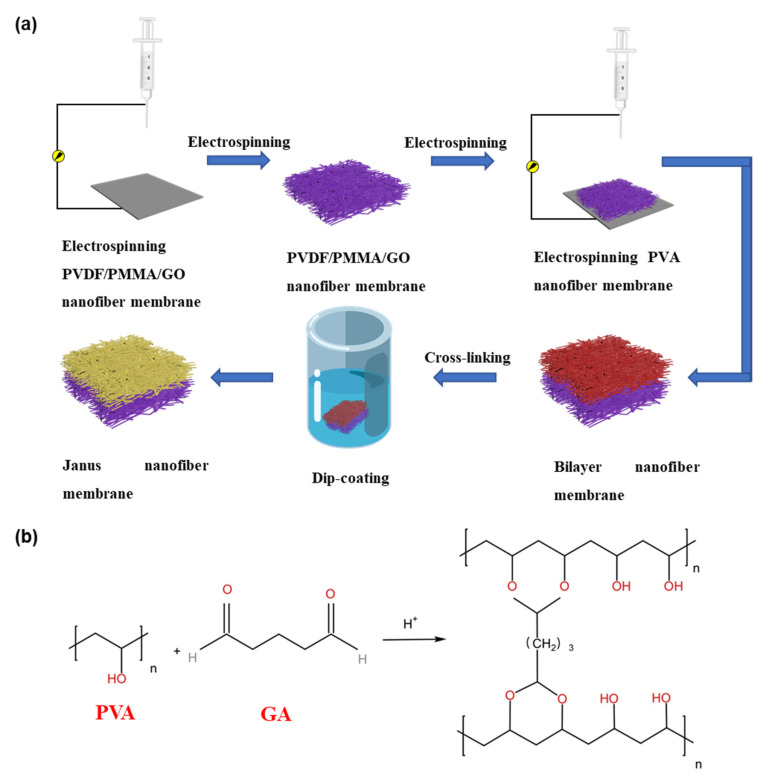
(**a**) Schematic diagram of the Janus membrane preparation process, (**b**) details of cPVA.

**Figure 2 polymers-14-00621-f002:**
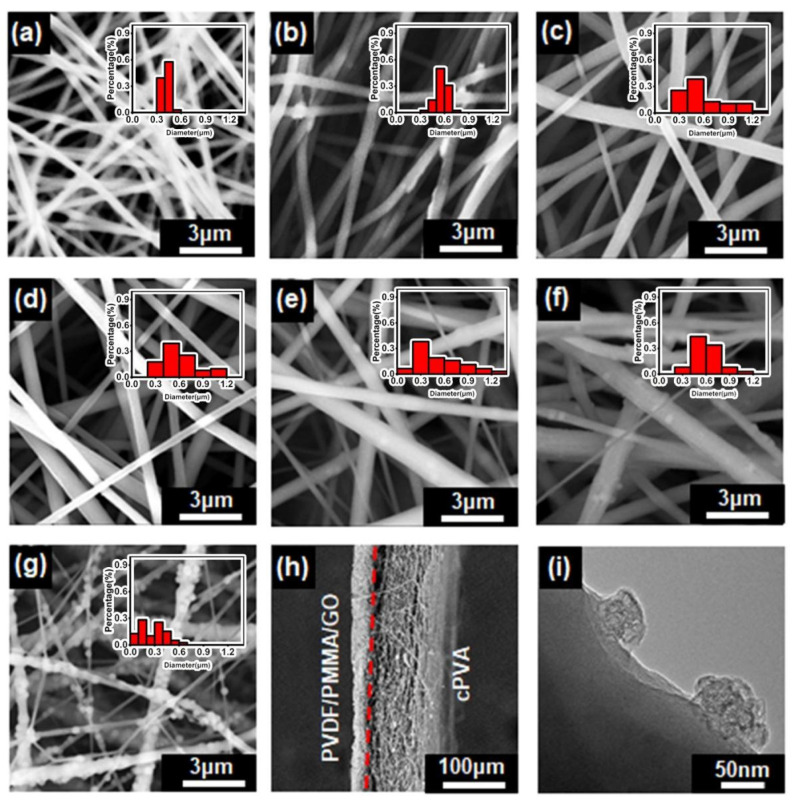
The SEM images of (**a**) PVA, (**b**) cPVA, (**c**) PVDF, (**d**) PVDF/PMMA, (**e**) PVDF/PMMA/0.8% GO, (**f**) PVDF/PMMA/1% GO, (**g**) PVDF/PMMA/1.2% GO, (**h**) cross section of Janus membrane cPVA-PVDF/PMMA/GO, (**i**) The TEM images of PVDF/PMMA/GO hydrophobic side of (**f**). The statistical distribution of fiber thickness was listed in the insert graphs in the upper right corner of (**a**–**g**).

**Figure 3 polymers-14-00621-f003:**
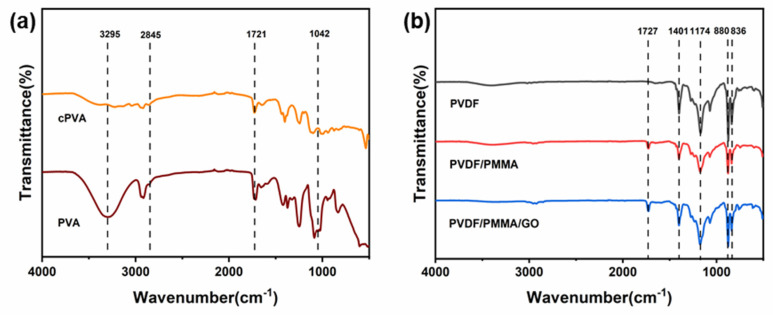
FTIR spectra of (**a**) Janus cPVA nanofiber membrane (hydrophilic side), (**b**) PVDF/PMMA/GO fiber membrane (hydrophobic side).

**Figure 4 polymers-14-00621-f004:**
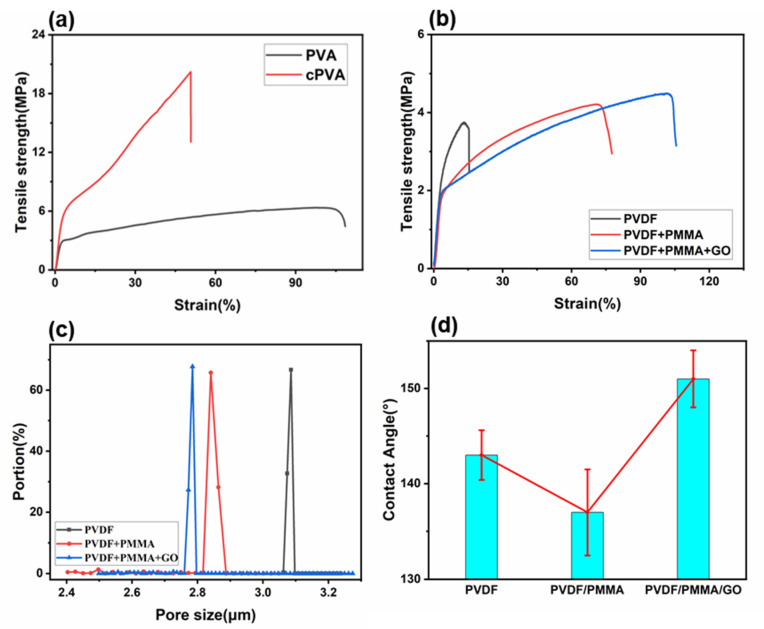
(**a**) Stress–strain curve of hydrophilic side, (**b**) stress–strain curve of hydrophobic side, (**c**) pore size distribution of hydrophobic layer, (**d**) comparison of contact angles of hydrophobic layers with different components.

**Figure 5 polymers-14-00621-f005:**
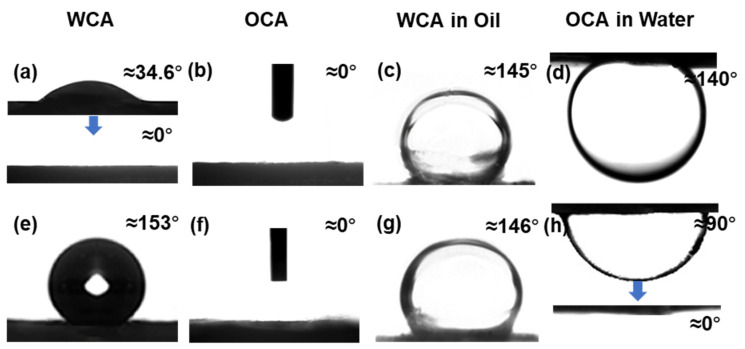
Surface wettability of the two active sides of the Janus membrane (**a**–**d**) cPVA side and (**e**–**h**) PVDF/PMMA/GO side.

**Figure 6 polymers-14-00621-f006:**
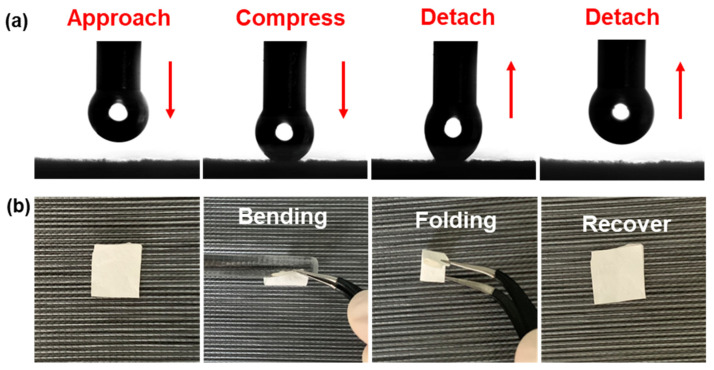
(**a**) Low adhesion on hydrophobic side (**b**) flexible Janus composite membrane.

**Figure 7 polymers-14-00621-f007:**
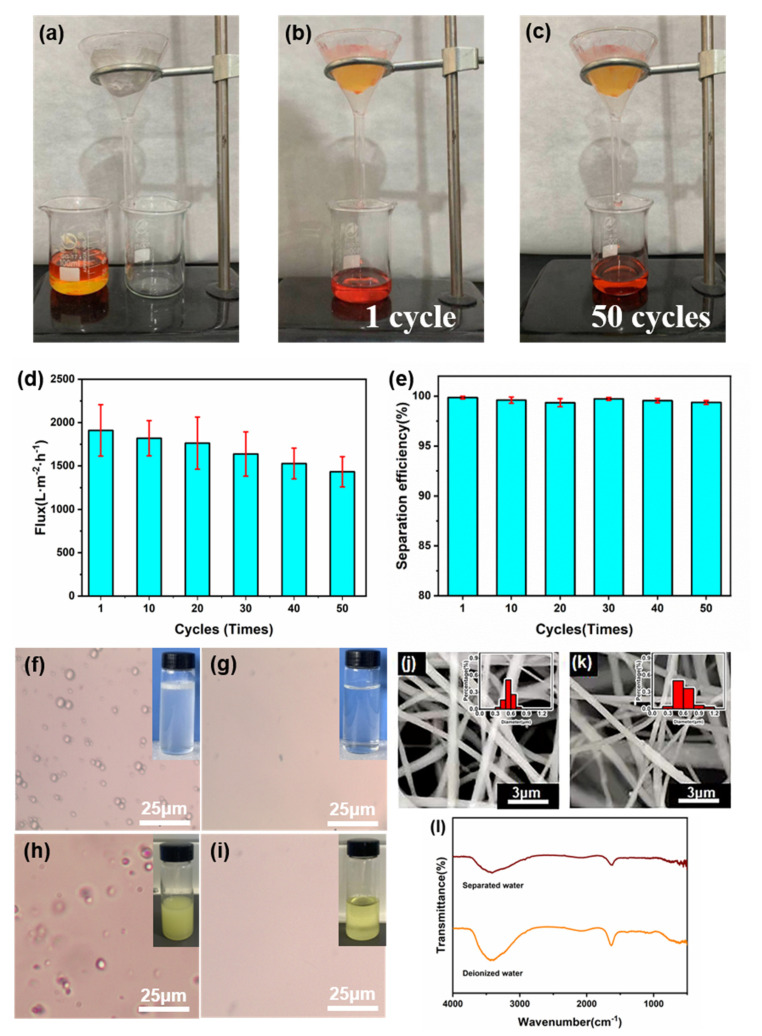
(**a**–**c**) Schematic diagram of oil-water separation experiment, (**d**) oil-water separation efficiency, (**e**) cyclic stability of the Janus membrane, (**f**,**g**) optical microscope images before and after filtration of water-in-oil emulsion, (**h**,**i**) optical microscope images before and after filtration of water-in-diesel emulsion, (**j**,**k**) SEM images of cPVA and PVDF/PMMA/GO after separation, (**l**) FTIR spectra of deionized water and separated water. The fiber diameter distribution is in the upper right corner of SEM.

**Figure 8 polymers-14-00621-f008:**
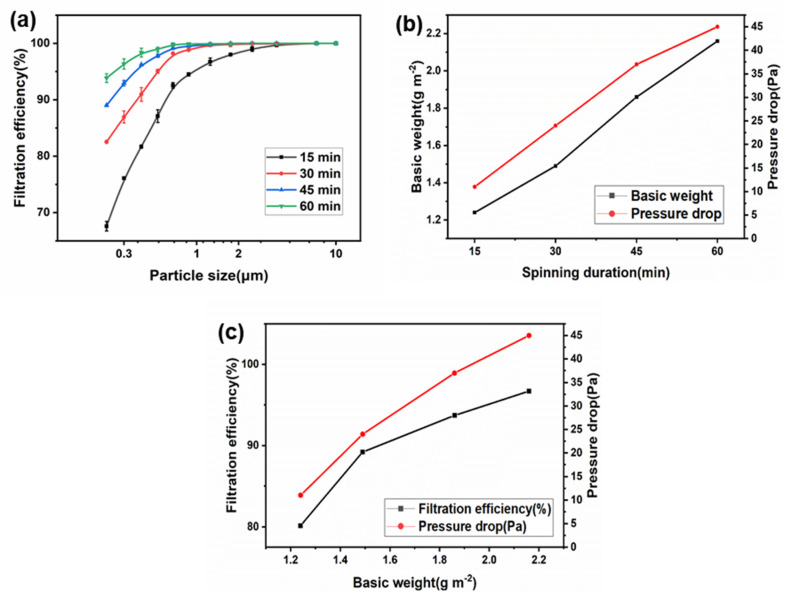
(**a**) Filtration efficiency curve of different hydrophilic layer spinning time (15–60 min), (**b**) relationship between spinning duration and basic weight and pressure drop, (**c**) relationship between basic weight and efficiency and pressure drop.

**Table 1 polymers-14-00621-t001:** Physical parameters of the water-in-diesel emulsions.

Property	Water-in-Diesel Emulsions
Viscosity (mPa × S)	3.52
Density (g cm^−3^)	0.82
Surface tension (mN m^−1^)	29.42
Water content (%)	5.8

**Table 2 polymers-14-00621-t002:** Wetting behavior of Janus composite membrane.

Membrane Side	θ_WA_		θ_OA_			θ_water–in–oil_		θo_il–in–water_	
	Contact Angle (°)	Wettability	Testing oil	Contact Angle (°)	Wettability	Contact Angle (°)	Wettability	Contact Angle (°)	Wettability
cPVA	0	Hydrophilic	n-hexane	0	Oleophilic	145	Hydrophobic	140	Oleophobic
PVDF/PMMA/GO	153	Hydrophobic	n-hexane	0	Oleophilic	146	Hydrophobic	0	Oleophilic

**Table 3 polymers-14-00621-t003:** Comparison of different preparation methods of Janus composite membrane.

Materials	Fabrication Method	Water Contact Angle of Hydrophobic Side	Major Applications	Reference
PLA-SiO_2_/PLA-CNTs	electrospinning	142°	oil–water separation	[53]
PVDF-M-CNT	coating	115.3°	membrane distillation	[52]
c-PVA/f-CNT	coating	157°	oil–water separation	[41]
PCFE/GO/JGO	coating	90°	water treatments	[54]
HP-PET/GNs-PET	coating	95.3°	oil–water separation	[55]
AuNR/SWNT	coating	110°	photothermal water desalination	[56]
CNTs/MPPM	coating	158°	oil–water separation	[57]
PAN/CNTs	coating	100°	oil–water separation	[44]
polydopamine-coated SWCNT/SWCNT	coating	104°	oil–water separation	[58]
cPVA/PVDF-PMMA-GO	electrospinning	153°	oil–water separation	This work

**Table 4 polymers-14-00621-t004:** Separation flux and efficiency of this study compared to other fibrous membranes.

Materials	Flux	Efficiency	Reference
PLA-SiO_2_/PLA-CNTs	1457 L m^−2^ h^−1^	99.1%	[53]
HP-PET/GNs-PET	≈1875 L m^−2^ h^−1^	99.0%	[55]
PNIPAM-PVDF/PVDF	1500 L m^−2^ h^−1^	96.0%	[32]
PAN/PS	430 L m^−2^ h^−1^	-	[60]
COF-DhaTab/PAN	1100 L m^−2^ h^−1^	99.9%	[61]
cPVA/PVDF-PMMA-GO	1910 L m^−2^ h^−1^	99.9%	This work

## Data Availability

Not applicable.

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
