# Peer review of "The High Flux of Superhydrophilic-Superhydrophobic Janus Membrane of cPVA-PVDF/PMMA/GO by Layer-by-Layer Electrospinning for High Efficiency Oil-Water Separation"

_polymers, 2022, doi:10.3390/polym14030621_

Round 1
Reviewer 1 Report
The paper refers to very important ecological problems. I suppose, that this work will be interesting for readers. I suggest only minor corrections:
- Please, provide in the Introduction the source of the name "Janus" origin.
- Please, correct indexes at units.
- Provide statistical distribution of fibers thickness at Figure 2.
- Line 234: it is mentioned that "....which can be attributed to the decrease of crystallinity of PVDF...". Did you measure crystallinity of the polymers?
- FTIR spectra of PVDF is shifted in comparison with the cited literature. Do you have an explanation?
- Since the work include GO additive, I suggest to modify following sentence and heat it up by additional reference, see below:
„ Electrospinning is a simple and widely used technology in the formation of micro-nanofibers. It can easily and effectively prepare one-dimensional (1D), two-dimensional (2D) and three-dimensional (3D) electrospun nanofibers based on various materials such as polymers, ceramics, composites, carbon materials and so on[1-3, DOI: 10.3390/POLYM12122766].
Author Response
Dear Editors and Reviewers:
Thank you for your letter and for the reviewers’ comments concerning our manuscript entitled “The High Flux of Superhydrophilic-superhydrophobic Janus Membrane of cPVA-PVDF/PMMA/GO by layer-by-layer Electrospinning for High Efficiency Oil-water Separation”. (ID: polymers-1573465). Those comments are all valuable and very helpful for revising and improving our paper, as well as the important guiding significance to our researches. We have studied comments carefully and have made corrections which we hope meet with your approval. Revised sentences are marked in yellow in the revised version. The main corrections in the paper and the responds to the Reviewers’ comments are listed as follows.
Once again, thank you very much for your comments and suggestions.
Best regards,
Jie Zheng, Dr.
Qingdao University,
No. 308, Ningxia Road,
Qingdao 266071, Shandong province, China
E-mail: [email protected]
Tel: +86 13792854370

Reviewer 2 Report
This article reports a simple and novel strategy for the preparation of hydrophilic –hydrophobic Janus membrane for high efficiency oil-water separation. The Janus membrane of cPVA-PVDF/PMMA/GO with opposite hydrophilic and hydrophobic properties was prepared by layer by-layer electrospinning. The work is interesting and applied nature from environmental and resources recycling point of view. However, the article needs further improvement in the light of following comments:
- The abstract should contain some numerical data in terms of efficiency of as-prepared materials for oil water separation.
- The novelity of this work should be addressed by writing comprehensive problem statement in the introduction section.
- The introduction part is mostly composed of the Janus membrane formation methods unnecessarily.
- There are some poorly written sentences in this manuscript. For example “Observe the functional groups of the nanocomposite membrane by Fourier transform infrared spectroscopy “.
- There should be recycling tests of cPVA-PVDF/PMMA/GO Janus membrane.
- There should be UV–Vis and FT-IR spectrophotometric evaluation of the separated water for contamination of the cPVA-PVDF/PMMA/GO Janus membrane. For guidance please read Journal of Molecular Liquids 290 (2019)111186.
- Physico-chemical characterization of separated water for reuse should be done.
- The as-prepared cPVA-PVDF/PMMA/GO Janus membrane should also be applied to real samples collected from oil fields.

Author Response

(The authors gave the same response as above.)

Round 2
Reviewer 2 Report
Dear Editor, I have gone through the revised version of the submitted manuscript. Though it is improved but some of my earlier comments need to be addressed. It is also desirable that the authors provide point to point reply to my earlier comments. Regards, Shah
Author Response

(The authors gave the same response as above.)

Round 3
Reviewer 2 Report
They need to apply their prepared cPVA-PVDF/PMMA/GO Janus membrane to at least one real sample collected from oil fields though they have tested the separation performance of the Janus membrane by using model sample of diesel oil emulsion.